# Verifying Out-of-Distribution Robustness in Multi-spectral Satellite Change Detection

## Abstract

Reliable multi-spectral change detection *on-board* satellites requires robustness under distribution shifts. We address this challenge from both the *certification* and *empirical* perspectives.

On the certification side, we adapt neural verification to the unique structure of change detection, accounting for sensor noise, encoder–decoder heads, and semantic evaluation. We introduce a *tail-tapped verifier* that transports input intervals to the final decoder tap and applies $\alpha$-CROWN solely to the decision head. This yields per-pixel logit-margin lower bounds, which we summarize through task-aligned predicates such as coverage, false positives, and minimum island size.

On the empirical side, we study out-of-distribution robustness across three representative backbones—U-Net style encoder–decoder (FresUNet), lightweight convolutional attention encoder–decoder (FALCONet), and transformer-inspired global attention encoder–decoder (AttU-Net)—on the Onera Satellite Change Detection (OSCD) dataset. We find that existing certificates vanish even for mild perturbations ($\varepsilon \geq 1/255$), while empirical robustness varies widely across architectures.

Our results highlight both the difficulty of certifying change detection and the promise of architecture design for achieving practical robustness. This establishes a foundation for principled verification and stress-tested deployment of satellite-based change detection models.

## 1 Introduction

Deploying multi-spectral change detection *on-board* satellites requires robustness to realistic test-time shifts that deviate from training data, including illumination and phenology changes, crop rotation, atmospheric/BRDF effects, and sensor quirks. In this setting, high in-distribution accuracy is not enough; operators need models whose *predictions* remain stable under small, physically meaningful perturbations, together with principled diagnostics for when this stability fails.

**Why this is hard.**  Formal certification in vision typically bounds $\ell_p$ neighborhoods around a fixed input Cohen et al. (2019); Salman et al. (2019). For deep encoder–decoders with skip connections, such bounds often explode as they propagate, especially in segmentation, leaving Ground Truth (GT)-aware certificates either vacuous or prohibitively loose. Meanwhile, standard augmentation and generic Out-of Distribution (OOD) benchmarks Yang et al. (2023); Hendrycks & Dietterich (2019) fail to capture sensor-calibrated perturbations in multi-spectral imagery, leaving a gap between training and what satellites actually observe.

**Our approach in a sentence.**  We propose to *train for stability where the decision is made* and to *diagnose why certificates fail*. Concretely, we introduce **Head-Consistency Training (HCT)**, a GT-aligned objective that enforces positive *head margins* under small, physically grounded perturbations such as illumination drift, shadows, blur, and passband shifts. In parallel, we design a lightweight *tail-tapped verifier* that transports intervals to the final decoder tap and applies a tight linear relaxation ($\alpha/\beta$-CROWN) only across the decision head Wang et al. (2021). The goal is not to claim strong certificates, but to expose where and why stability breaks.

**What we find.** Across representative backbones on Onera Satellite Change Detection (OSCD) Daudt et al. (2018b) dataset and a curated crop-rotation OOD set, HCT improves *empirical* segmentation quality, both on clean inputs and under small synthetic shifts. Yet *GT-aware certificates remain scarce* for $\varepsilon \geq 1/255$. Our diagnostic consistently shows *tight head logit spans but rapidly widening encoder intervals at the tap* as $\varepsilon$ grows—explaining certificate collapse even when predictions still appear plausible.

**Contributions.** Our main contributions are:

1. **Head-Consistency Training (HCT):** a simple, GT-aligned head-margin objective under sensor-calibrated perturbations that yields stronger empirical robustness without adversarial inner loops.

2. **Physically grounded OOD protocol:** four Sentinel-2–motivated perturbation families (low-frequency drift, shadowing, passband shift, mild blur) and a real crop-rotation benchmark, both plug-and-play for multi-spectral CD.

3. **Verification as diagnosis:** a head-only, tail-tapped $\alpha/\beta$-CROWN verifier that reveals the head–tail tightness gap (encoder widening vs. head span), guiding architectural and training changes.

4. **Takeaway:** with current encoder–decoders, empirical gains from HCT do not translate into certified robustness at small budgets; narrowing encoder bounds (e.g., verification-friendly pooling/upsampling and skip control) appears the most promising route forward.

**A quick look at results.** *Diagnostics:* On OSCD, strict GT-aware certificates vanish for $\varepsilon \geq 1/255$ across FresUNet, FALCONet, and AttU-Net, while our tap-level statistics show that head spans remain tight but encoder/tap intervals widen sharply with $\varepsilon$ (Table 1, left). *HCT on OSCD:* Head-Consistency Training raises clean performance for a lightweight backbone to **F1** = 0.624 with **precision** = 0.536 and **recall** = 0.746 (the highest reported clean accuracy on OSCD - Table 1, right). *Semantic OOD:* The same model transfers to the curated crop-rotation benchmark with **Dice** = 0.219 (**precision** = 0.238, **recall** = 0.192), substantially above non-HCT baselines (Table 1, right).

Taken together, the verifier and its diagnostics expose *why* certificates fail at realistic budgets, while HCT and our synthetic/semantic OOD resources provide a practical path to *empirical* robustness and transparent auditing. We release training code (HCT), perturbation generators, and the diagnostic verifier, and hope this combination—diagnostic certification, lightweight robustness training, and reproducible OOD protocols—serves as a compact, repeatable baseline for multi-spectral change detection.

## 2 RELATED WORK

**Neural network verification.** Convex relaxations via LiRPA (e.g., CROWN/$\alpha$-CROWN) provide scalable certified bounds by propagating linear relaxations layerwise Wang et al. (2021). Hybrids such as CROWN-IBP use interval bounds as warm-up, then refine them with linear relaxations, achieving strong results in VNN-COMP Brix et al. (2023). AutoLiRPA Xu et al. (2020) has made these methods practical. Our work follows this line but *restricts* verification to the semantic head: we transport uncertainty to a decoder tap and apply $\alpha/\beta$-CROWN locally. This places us in partial-verification methods, trading global coverage for tight, task-relevant diagnostics.

**Certificates for dense prediction.** Robustness for segmentation has been studied via randomized smoothing Cohen et al. (2019); Salman et al. (2019) and LiRPA variants that aggregate pixelwise evidence or impose structural priors Fischer et al. (2021); Kumar et al. (2021); Hao et al. (2022). These succeed when score maps remain well separated and abstraction error is controlled, but for encoder–decoders with skip paths, end-to-end relaxations quickly become vacuous. Our head-tapped verifier addresses this gap by evaluating *GT-aligned predicates* (overlap, FP, structure) that make explicit why certificates collapse: widening encoder intervals despite stable head spans.

**Satellite CD backbones.** Early CD models in multi-spectral remote sensing used *early fusion* (stack $x_1, x_2$) or *Siamese* designs with weight sharing Daudt et al. (2018a;b). In practice, encoder–decoder segmenters dominate: U-Net Ronneberger et al. (2015) and attention-gated U-Net Oktay et al. (2018) remain defaults under small datasets and memory budgets, while DeepLabv3+ Chen et al.

(2018) is often adapted by changing the head to two logits. These backbones are attractive for on-orbit use since they (i) support arbitrary band counts, (ii) expose a simple $1 \times 1$ head where decisions are made, and (iii) train patch-wise on OSCD Daudt et al. (2018b). Our experiments therefore span three families—plain U-Net, attention-gated U-Net, and a lightweight convolutional-attention variant—plus a DeepLabv3+ baseline.

**Robust change detection and OOD.** Recent Sentinel-2 CD models build on U-Net variants with attention or lightweight modules, or on remote sensing foundation models Hong et al. (2024). Robustness has been addressed mainly by augmentation, adversarial perturbations, or cross-domain transfer Yang et al. (2023); Hendrycks & Dietterich (2019). Existing OOD suites rarely capture sensor-calibrated perturbations or semantic shifts such as crop rotation. We contribute both: (i) structured perturbation families aligned with sensor physics and (ii) a curated CropRot benchmark, designed to stress-test models beyond generic OOD.

**Training for robustness.** Verifier-aware training (IBP, CROWN-IBP) mixes cross-entropy with bound losses to enlarge certified margins Wang et al. (2021). Our Head-Consistency Training (HCT), by contrast, is *not* adversarial and *not* verifier-coupled: it enforces margin preservation at the head under sensor-calibrated perturbations. HCT improves empirical robustness but—consistent with our diagnostics—does not close the certification gap when encoder intervals remain wide. Bridging that gap likely requires tighter tap bounds (in-graph normalization, certified pooling/upsampling, skip-path control) together with stronger head relaxations ($\beta$-CROWN, cut-plane tightening) or shallow-body verification.

## 3 METHOD

### 3.1 TAIL-TAPPED $\alpha$-CROWN FOR CERTIFIED CHANGE SEGMENTATION

We seek guarantees that the *semantic decision* (change vs. no-change) made by a change-detection model remains stable under bounded, structured perturbations such as illumination shifts, mild occlusions, or crop-type variation. Rather than certifying all layers end-to-end, we certify only where the decision is made: the final semantic head (DoubleConv $\rightarrow 1 \times 1$). Our verifier transports input uncertainty to the last decoder features using interval arithmetic and then applies a tight $\alpha$-CROWN relaxation *only* on this tail. The resulting per-pixel certified margins are aggregated into interpretable, task-level predicates.

#### 3.1.1 PROBLEM SETUP AND CERTIFIED MARGIN

Let a trained model map a co-registered pair $(x_1, x_2)$ to per-pixel logits

$$z = f(x_1, x_2) \in \mathbb{R}^{C \times H \times W},$$

with channels $(c_{\text{chg}}, c_{\text{nchg}})$ corresponding to change and no-change. We consider perturbations $(\delta_1, \delta_2)$ from a bounded set $\Delta_\varepsilon$ (e.g., $\ell_\infty$ balls of radius $\varepsilon$ with content-preserving transforms). For pixel $p$, the binary margin is

$$m_p(x_1, x_2) = z_{c_{\text{chg}}}(p) - z_{c_{\text{nchg}}}(p).$$

A pixel is *certified* at radius $\varepsilon$ if a sound lower bound $m_p^{\text{lb}}(x_1, x_2, \varepsilon) > 0$ is established, implying class invariance for all admissible perturbations.

#### 3.1.2 BACKBONES AND OSCD PROTOCOL

**OSCD setup.** We use the Onera Satellite Change Detection (OSCD) dataset Daudt et al. (2018b), which provides co-registered Sentinel-2 image pairs and pixel-level change masks for multiple cities. Inputs are pairs $(x_1, x_2)$ stacked channel-wise, yielding 26 channels ($2 \times 13$). Unless stated otherwise, models produce logits $z \in \mathbb{R}^{2 \times H \times W}$ for change/no-change. Our main experiments use five diverse cities (`brasilia`, `lasvegas`, `dubai`, `paris`, `abudhabi`) and perturbation radii $\varepsilon \in \{0, 1/255, 2/255\}$.

**Block library.** We rely on standard building blocks:

- `DoubleConv`: two successive $3 \times 3$ convolutions (with nonlinearity and optional normalization).

- Up: an upsampling step (bilinear or transposed convolution) followed by DoubleConv, typically concatenating a skip feature.

- OutConv: a $1 \times 1$ convolution mapping decoder features to $C$ logits per pixel.

- Attention gate / MHA: lightweight modules for modulating skips or injecting context.

**Three representative backbones.**

1. **FresUNet:** a vanilla U-Net with symmetric DoubleConv blocks, skip concatenations, and an OutConv head.

2. **FALCONet:** a U-Net trunk augmented with convolutional and lightweight multi-head attention, improving local context while retaining the same decoder/head interface.

3. **AttU-Net:** an attention-gated U-Net where skip connections are modulated before concatenation, with a standard decoder and OutConv head.

All three produce logits $z \in \mathbb{R}^{2 \times H \times W}$ with identical channel semantics, enabling a uniform verification interface. Complete layer specifications and the body/tail split are given in Appendix G.

### 3.1.3 BODY/TAIL FACTORIZATION AND THE TAIL TAP

We decompose each backbone into a *body $B$* and a *tail $T$*:

$$f(x_1, x_2) = T\big(B(x_1, x_2)\big).$$

The body $B$ is the encoder–decoder trunk up to the final skip concatenation; the tail $T$ is the last DoubleConv followed by the $1 \times 1$ head. This decomposition is identical across backbones and defines a uniform hook point for certification.

**Interval transport.** We propagate the input $\ell_\infty$ box through $B$ using interval arithmetic to obtain

$$z \in [\ell, u] := \big[\underline{z}(B, \varepsilon), \, \overline{z}(B, \varepsilon)\big].$$

This step is deterministic and cheap, but interval bounds alone are loose: they tend to inflate pre-logit ranges through deep decoders, making head certificates vacuous (lower bounds $\leq 0$) or unstable in coverage (Appendix C).

### 3.1.4 $\alpha$-CROWN ON THE TAIL

On the box $[\ell, u]$, the tail $T$ consists of a short sequence of convolutions, biases, nonlinearities, and a $1 \times 1$ head. $\alpha$-CROWN builds per-layer linear relaxations and composes them into a global affine lower bound on the pixelwise margin:

$$m_p\big(T(z)\big) \; \geq \; a_p^\top z + b_p, \qquad \forall z \in [\ell, u],$$

so that $m_p^{\mathrm{lb}}(\ell, u) = \min_{z \in [\ell, u]}(a_p^\top z + b_p)$. Because $B$ is abstracted by $[\ell, u]$, optimization is confined to the short tail, yielding tight bounds at low cost.

### 3.1.5 CERTIFIED PIXEL SETS AND SEMANTIC PREDICATES

Given bounds $\underline{z}, \overline{z} \in \mathbb{R}^{C \times H \times W}$ at radius $\varepsilon$, fix change channel $c_{\mathrm{chg}}$ and no-change channel $c_{\mathrm{nchg}}$. For pixel $p$, define the certified margin lower bound

$$\underline{m}(p) = \underline{z}_{c_{\mathrm{chg}}}(p) - \overline{z}_{c_{\mathrm{nchg}}}(p), \quad \mathcal{C}_{\mathrm{cert}} = \{p : \underline{m}(p) > 0\}.$$

With clean-prediction set $\mathcal{C}_{\mathrm{clean}}$ and ground truth $\mathcal{C}_{\mathrm{gt}}$, we define:

$$\textbf{Overlap:} \quad \mathsf{P}_{\mathrm{overlap}}(\rho) = \left[ \frac{|\mathcal{C}_{\mathrm{cert}} \cap \mathcal{C}_{\mathrm{clean}}|}{\max(1, |\mathcal{C}_{\mathrm{clean}}|)} \geq \rho \right],$$

$$\textbf{False Positives:} \quad \mathsf{P}_{\mathrm{fp}}(\gamma) = \left[ \frac{|\mathcal{C}_{\mathrm{cert}} \setminus \mathcal{C}_{\mathrm{gt}}|}{\max(1, |\mathcal{C}_{\mathrm{cert}}|)} \leq \gamma \right],$$

$$\textbf{Pattern:} \quad \mathsf{P}_{\mathrm{pattern}}(s_{\min}) = \bigwedge_k \big[ |S_k| \geq s_{\min} \big] \quad \text{for 4-connected components } \{S_k\} \text{ of } \mathcal{C}_{\mathrm{cert}}.$$

We report the strict conjunction $P_{\text{overlap}} \wedge P_{\text{fp}} \wedge P_{\text{pattern}}$. These predicates elevate per-pixel certificates into semantic guarantees on overlap, precision, and structural consistency. Detailed algorithms, soundness guarantees, and implementation are provided in Appendix B.

Full verifier proofs, predicate algorithms, and backbone specifications are provided in Appendix B.

## 3.2 HEAD-CONSISTENCY TRAINING (HCT) FOR PHYSICALLY GROUNDED OOD

**Goal.** We aim to make the *decision head* (change vs. no-change) stable under small, sensor-calibrated shifts common in Sentinel-2 imagery—illumination drift, shadows, mild blur, and pass-band differences. Unlike adversarial training, HCT has *no inner maximization*. Instead, we sample physically motivated perturbations and enforce that the head preserves the *ground-truth* (GT) label under these shifts.

**Setup.** A model $f$ maps a co-registered pair $(x_1, x_2)$ to logits

$$z = f(x_1, x_2) \in \mathbb{R}^{2 \times H \times W}, \quad \{0 = \text{no-change}, 1 = \text{change}\}.$$

For each training pair, we draw $K$ independent transforms $\mathcal{S}_\varepsilon^{(k)}$ from a sensor-calibrated family (budget $\varepsilon$, Sec. 3.3), form perturbed inputs $\tilde{x}_i^{(k)} = \mathcal{S}_\varepsilon^{(k)}(x_i)$, and compute logits $\tilde{z}^{(k)} = f(\tilde{x}_1^{(k)}, \tilde{x}_2^{(k)})$.

**GT-aligned head-margin consistency.** Let $y(p) \in \{0, 1\}$ be the GT label at pixel $p$ and $\bar{y}(p) = 1 - y(p)$. For perturbation $k$, define the GT margin

$$m^{(k)}(p) = \tilde{z}_{y(p)}^{(k)}(p) - \tilde{z}_{\bar{y}(p)}^{(k)}(p).$$

We penalize shortfalls below a margin buffer $\tau$:

$$\mathcal{L}_{\text{HCT}} = \frac{1}{KHW} \sum_{k=1}^{K} \sum_{p} \phi\big(\tau - m^{(k)}(p)\big),$$

with $\phi(u) = \max(0, u)$ (hinge) or $\phi(u) = \log(1 + e^u)$ (softplus). We set $\tau \in [0.05, 0.15]$ logits.

**Supervised fit and optional regularizers.** We retain standard cross-entropy on clean inputs and optionally add Dice for rare positives:

$$\mathcal{L}_{\text{CE}} = \text{CE}(z, y), \qquad \mathcal{L}_{\text{Dice}} = 1 - \frac{2 \langle \sigma(z_1), y \rangle + \epsilon}{\|\sigma(z_1)\|_1 + \|y\|_1 + \epsilon}.$$

The total loss is

$$\mathcal{L} = \lambda_{\text{CE}} \mathcal{L}_{\text{CE}} + \lambda_{\text{HCT}} \mathcal{L}_{\text{HCT}} + \lambda_{\text{Dice}} \mathcal{L}_{\text{Dice}} \text{ (optional)},$$

with $\lambda_{\text{CE}} = 1$, $\lambda_{\text{HCT}} \in [0.2, 0.5]$, $\lambda_{\text{Dice}} \in [0, 0.2]$.

**Practical recipe.** Each minibatch contains one clean copy and $K \in \{1, 2\}$ perturbed copies (default $K=1$) sharing the GT. We ramp $\varepsilon$ linearly from 0 to $\varepsilon_{\max}$ over the first 30–40% of epochs, then hold it fixed. Other practical notes:

- Normalize inside the graph (per-channel standardization).
- Use ReLU/LeakyReLU; avoid unstable `ConvTranspose2d` settings.
- Apply weight decay $10^{-4}$–$5 \times 10^{-4}$, gradient clip 1.0, and optionally EMA ($\leq 0.999$).
- Class weighting or focal loss may be added if positives are extremely sparse.

## 3.3 PHYSICALLY GROUNDED SYNTHETIC OOD PERTURBATIONS

We operate in normalized 8-bit space, where $\varepsilon = 1/255 \approx 0.0039$ corresponds to a $\sim 0.39\%$ per-band reflectance change. We sample an equal mix of four perturbation families, used both in HCT training (Sec. 3.2) and for empirical robustness evaluation (Table 1, right).

1. **lf_1, lf_2 (low-frequency drift).** Smooth additive fields $\delta(x, y)$ drawn by Gaussian-filtered white noise (broad bandwidth for lf_1, narrower for lf_2), applied per band with $\|\delta\|_\infty \leq \varepsilon$.

2. **shadow.** Soft multiplicative vignette $v(x, y) \in [1-\varepsilon,\, 1+\varepsilon]$, created from a cosine ramp along a random direction and smoothed; mimics cast/self shadows and BRDF effects.

3. **pband (passband shift).** Per-band affine change $x_b \mapsto (1 + \alpha_b)x_b + \beta_b$, with $|\alpha_b| \leq \varepsilon$, $|\beta_b| \leq \varepsilon/2$, approximating inter-sensor spectral response differences.

4. **blur_1.** Mild Gaussian blur with $\sigma \in [0, 1.0]$ pixels, capped to keep induced $\ell_\infty$ per-band change below $\varepsilon$.

**Why these?** All four perturbations are common in Sentinel-2 time series and realistic at $\varepsilon \in \{1, 2\}/255$. They probe whether the decision head preserves semantics under small radiometric/optical shifts, complementing certification diagnostics with empirically grounded stress tests.

### 3.4 REALISTIC OOD PROTOCOL (CROPROT)

We introduce **CropRot**, a curated OOD protocol that emphasizes vegetation-driven change in Sentinel-2 imagery. Masks are derived by differencing Normalized Difference Vegetation Index (NDVI) maps across time after cloud-screened pair selection. NDVI differencing is long used in remote sensing Tucker (1979); Singh (1989); Coppin et al. (2004), but directly thresholding $\Delta$NDVI can capture phenology, BRDF/atmospheric variation, or misregistration rather than semantic change. To mitigate this, we add a lightweight *visual quality assurance* step: per scene, we apply false-color composites and $\Delta$NDVI heatmaps to verify change patterns and, when needed, adjust the threshold within a narrow range. This yields *semi-automatic, curated* masks that are reproducible (scripts emit file lists, filters, and chosen thresholds) yet remain proxies rather than full manual annotations.

#### 3.4.1 CURATED MASKS VIA NDVI DIFFERENCING AND QA.

To generate weak semantic change labels, we start with cloud-free Sentinel-2 pairs $(T_1, T_2)$ retrieved via the SentinelHub API (Level-2A surface reflectance, 13 bands). NDVI is computed as

$$\mathrm{NDVI}(x, y) = \frac{B_8(x, y) - B_4(x, y)}{B_8(x, y) + B_4(x, y)},$$

with $B_4$ (red) and $B_8$ (NIR). Differencing across time gives

$$\Delta\mathrm{NDVI}(x, y) = \mathrm{NDVI}_{T_2}(x, y) - \mathrm{NDVI}_{T_1}(x, y).$$

A threshold $\theta$ binarizes the map:

$$\mathrm{Change}(x, y) = \mathbb{1}\big(\Delta\mathrm{NDVI}(x, y) > \theta\big).$$

Visual QA overlays (spectral color ramps, $\Delta$NDVI heatmaps) ensure threshold choice matches observable vegetation dynamics. Figure 1 summarizes the pipeline.

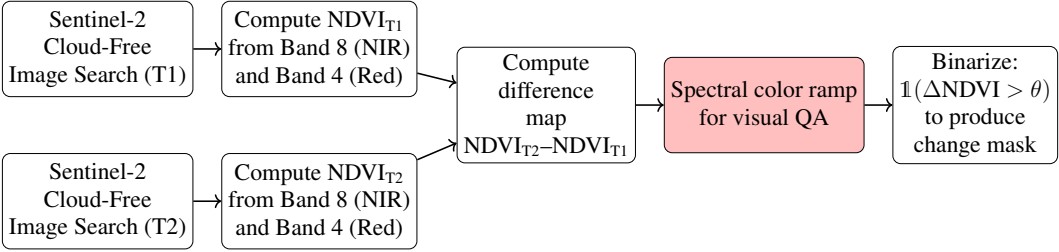

Figure 1: Pipeline for generating weak change masks via NDVI differencing. Cloud-free Sentinel-2 images are retrieved, NDVI maps computed, differenced, and thresholded with visual QA.

**OOD emphasis.** Figure 2 contrasts OSCD (urban change) and CropRot (vegetation change). Both share identical 13-band interfaces; CropRot stresses temporal vegetation dynamics and thus provides a semantically distinct OOD benchmark.

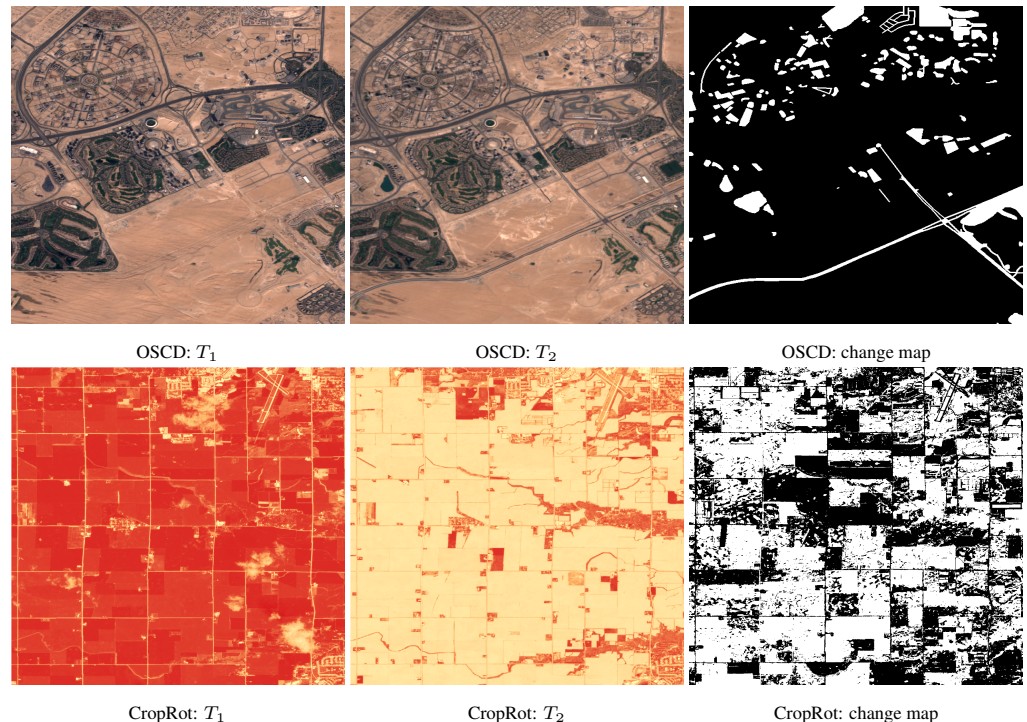

Figure 2: **OSCD vs. CropRot (OOD).** OSCD emphasizes urban change; CropRot emphasizes vegetation change. Both share the same Sentinel-2 interface, making CropRot a natural OOD benchmark.

**Sanity check.** To assess dataset quality, we trained standard architectures with OSCD-like hyperparameters. Models converged quickly, reaching F1 comparable to OSCD. For example, a simple encoder–decoder achieved F1=0.518 on OSCD and F1=0.526 on CropRot. This indicates CropRot is sufficiently clean for modeling. Its intended role, however, is not as a primary training set but as a realistic OOD benchmark for models trained on OSCD. In addition to the released CropRot benchmark, we have curated several further candidate regions ($\sim 8$ sites in Ukraine and Occitanie). These were excluded from the present release due to crop-type ambiguity or barren fields. We plan to polish and include them in future versions of the dataset.

**Note on reproducibility.** We provide full curation details in Appendix A so that CropRot can be exactly re-generated and, more importantly, so that similar OOD benchmarks can be constructed for other regions or sensing modalities. CropRot should be viewed not as a large-scale dataset but as a *sample protocol* demonstrating how reproducible weak supervision can yield realistic OOD evaluation resources.

## 4 RESULTS

### 4.1 EXPERIMENTAL SETUP

**Main backbones.** We evaluate three certified backbones: (i) FresUNet Daudt (2020), a U-Net baseline (1.10M params; ∼524M mult-adds), (ii) FALCONet, a local convolution+attention variant (1.15M params; ∼526M mult-adds), and (iii) AttU-Net Oktay et al. (2018), an attention-gated U-Net (34.89M params; ∼16.8G mult-adds).

**Empirical baselines.** For context we also report in-distribution metrics on two large backbones: SpectralGPT+ Hong et al. (2024) (0.11B params; ∼168G mult-adds) and DeepLabv3+ Schmitt et al. (2020) (40.4M params; ∼332G mult-adds). These highlight accuracy vs. robustness trade-offs but are excluded from certification due to scale.

**Training and evaluation.** Metrics follow Daudt (2020): Precision (Change), Recall, F1, Dice, and Kappa (agreement beyond chance, important for OSCD imbalance). Models are trained with weighted NLL loss (inverse class frequency) plus an optional Dice term, optimized using Adam ($10^{-3}$ lr, $10^{-4}$ weight decay) with exponential decay (0.95). We train for 50 epochs with batch size 8 and $128 \times 128$ patches, selecting the best checkpoint by validation F1.

**Compute.** Most experiments (including certification) run on a 16 GB RAM laptop (PyTorch 2.x). Larger baselines and HCT training use a 128 GB RAM workstation with a 48 GB GPU.

## 4.2 CERTIFIED CHANGE SEGMENTATION

Across all three backbones, GT-aware certification collapses at small budgets ($\varepsilon \geq 1/255$) despite plausible clean maps. Table 1 combines *diagnostics* (tap interval width vs. head logit span) with *predicate outcomes*. We find:

- Head logit spans remain essentially constant. - Tap widths explode with $\varepsilon$, enlarging the uncertainty box. - As a result, certified margins vanish and $\mathcal{C}_{\text{cert}}$ shrinks to zero—even when clean predictions look reasonable.

Formally, with affine bound $m(z) \geq a^\top z + b$ (from $\alpha$-CROWN),

$$m_{\text{lb}}(\ell, u) \ \geq \ m_{\text{clean}} - \tfrac{1}{2} \sum_i |a_i|(u_i - \ell_i), \quad m_{\text{clean}} = a^\top \tfrac{\ell+u}{2} + b. \tag{1}$$

Thus margins deteriorate as tap width $(u-\ell)$ grows or head sensitivity $\|a\|_1$ increases. Appendix F formalizes this diagnosis.

### 4.2.1 PRACTITIONER TAKEAWAYS

**Network design.** Favor operations that keep tap boxes tight and head sensitivity moderate: average pooling, $1 \times 1$ bottlenecks, skip gating, in-graph normalization, near-1-Lipschitz activations. Avoid unstable `ConvTranspose2d` and exotic activations.

**Verification stack.** Reduce over-estimation by using tighter abstract domains (zonotopes, DeepPoly, $\beta$-CROWN with bound tightening), and add native relaxations for pooling/upsampling/attention. Where hotspots persist, lightweight local branch-and-bound at the head can recover certificates.

**Bottom line.** Two levers matter: (i) architecturally narrow the tap interval, and (ii) algorithmically tighten relaxations. Coupled with stability-aware training, these offer a pragmatic path toward certifiable on-board change detection.

## 4.3 EMPIRICAL OOD SEGMENTATION

We report Precision/Recall/Dice (P/R/D) on OSCD under synthetic corruptions and on CropRot (OOD).[1]

Key results (Table 1): - **Clean OSCD:** FALCONet[HCT] improves to Dice 0.62. AttU-Net remains strong. - **Synthetic OOD:** All models degrade under LF/shadow/pband/blur. HCT stabilizes but does not eliminate failures. - **CropRot (real OOD):** AttU-Net transfers best (Dice $\approx$0.35). HCT lifts FALCONet substantially (Dice 0.22 vs. 0.01), narrowing the gap. FresUNet collapses.

**Qualitative evidence.** Fig. 3 shows OSCD `brasilia`. AttU-Net captures dense, GT-aligned islands; FALCONet under-covers small parcels, consistent with its lower CropRot Dice.

## 4.4 COMPUTE FOOTPRINT: HCT VS. ADVERSARIAL TRAINING

HCT introduces no inner maximization and uses only one perturbed copy per minibatch. In practice, a lightweight model trains for $\sim$60 epochs in $\sim$3 hours on a 16 GB RAM CPU laptop. By contrast, adversarial training requires multi-step PGD per batch and typically a GPU, significantly increasing runtime and energy. This makes HCT a practical knob for improving *empirical* OOD behavior even when formal certificates remain elusive.

---

[1] *LF* = low-frequency drift; *shadow* = multiplicative shading map; *pband* = passband shift; *blur_1* = mild blur.

Table 1: **Left:** Head–tail diagnostics on OSCD (`width_mean` vs. head logit span). **Right:** Empirical OOD P/R/D.

**(a) Diagnostics**

| Model | $\varepsilon$ | Tap `width_mean` (med / p95) | Head logit span (med / p95) |
|---|---|---|---|
| FresUNet | 0/255 | 0/0 | 6.09/6.09 |
| | 1/255 | $1.58\times10^{7}/1.58\times10^{7}$ | |
| | 2/255 | $3.26\times10^{7}/3.26\times10^{7}$ | |
| FALCONet | 0/255 | 0/0 | 7.03/7.03 |
| | 1/255 | $3.17\times10^{4}/3.17\times10^{4}$ | |
| | 2/255 | $5.42\times10^{4}/5.42\times10^{4}$ | |
| AttU-Net | 0/255 | 0/0 | 7.49/7.49 |
| | 1/255 | $5.93\times10^{8}/5.93\times10^{8}$ | |
| | 2/255 | $1.17\times10^{9}/1.17\times10^{9}$ | |

**(b) OOD segmentation (P/R/D)**

| Model | OSCD clean | OSCD LF/shadow/ pband/blur | CropRot OOD |
|---|---|---|---|
| FresUNet | 0.47/0.58/0.52 | 0.00 | 0.11/0.00/0.00 |
| FALCONet | 0.55/0.63/0.59 | 0.00 | 0.52/0.01/0.01 |
| AttU-Net | 0.59/0.60/0.59 | 0.04/0.53/0.07 | **0.37/0.33/0.35** |
| FALCONet$^{\text{HCT}}$ | **0.54/0.75/0.62** | **0.05/0.02/0.03** | 0.24/0.19/0.22 |
| SpectralGPT | 0.16/0.50/0.24 | 0.06/0.23/0.04 | 0.13/0.18/0.11 |
| DeepLabv3 | 0.58/0.39/0.28 | 0.12/0.31/0.12 | 0.26/0.12/0.14 |

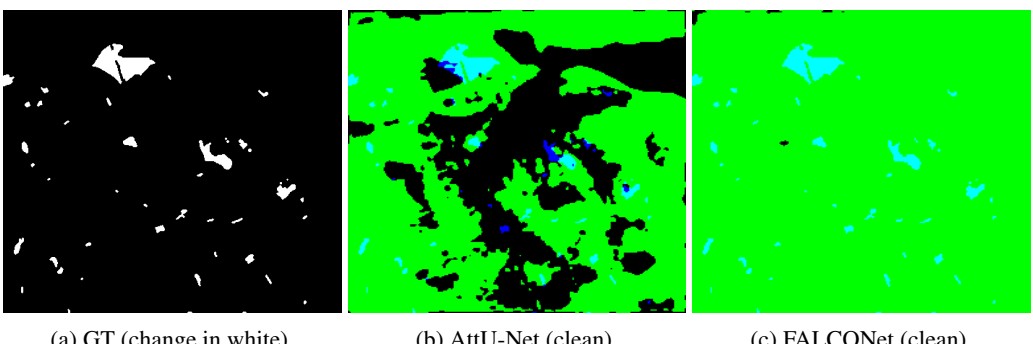

(a) GT (change in white).    (b) AttU-Net (clean).    (c) FALCONet (clean).

Figure 3: **OSCD `brasilia` (clean).** AttU-Net recovers denser GT-like change islands. This structure aligns with its stronger CropRot transfer (Table 1, right).

## 5 CONCLUSION

Robust certification remains especially difficult for skip-heavy encoder–decoders, which are essential to capture fine-grained spatial and spectral structure in multi-spectral imagery. Our verifier, limited to $L_\infty$ radiometric perturbations, fails at small budgets because encoder bounds widen excessively; head spans remain tight, but the bottleneck lies in the body. Head-Consistency Training (HCT) improves empirical robustness but does not make models certifiably robust. The CropRot dataset, while valuable as a vegetation-driven OOD protocol, is geographically narrow and should be expanded across regions and seasons.

Despite these limitations, our study contributes two complementary lenses: (i) a head-centric *diagnostic verifier* that makes explicit *why* formal guarantees collapse, and (ii) a lightweight training recipe—**HCT**—that stabilizes empirical predictions under sensor-calibrated shifts. Across backbones, certified guarantees vanish already at $\varepsilon \geq 1/255$ due to exploding encoder intervals, yet HCT lifts a compact FALCONet from Dice $\approx 0.01$ to **0.22** on the CropRot OOD benchmark while maintaining **0.62** on OSCD, narrowing the gap to AttU-Net ($\approx 0.35$ Dice). Synthetic corruptions remain challenging, underscoring the gap between clean accuracy and robust transfer.

**Takeaway and Outlook.** HCT offers a *practical middle ground* for on-board deployment: head-focused, GT-aligned, and efficient (no adversarial inner loops), trainable on modest hardware. Operators can pair compact backbones with HCT for deployment, while using our diagnostic verifier and OOD protocols for transparent auditing. Looking forward, progress likely requires: (i) tighter encoder bounds (native relaxations, $\beta$-CROWN, cut-plane tightening), (ii) structured-shift certification beyond $L_\infty$ (illumination, passband families), and (iii) training–verification co-design coupling HCT with bound-aware regularization. Together, these steps could narrow the gap between empirical OOD robustness and certifiable guarantees for complex, skip-heavy encoder–decoders in multi-spectral change detection.

We use LLMs to refine our text, i.e., for linguistic polish and grammatical accuracy.

**Reproducibility Statement.** We release training code for HCT, perturbation generators, and the diagnostic verifier to support transparent replication of all results. The CropRot OOD dataset is described in Section 3.4 and detailed curation steps are provided in Appendix A. Verifier proofs, predicate algorithms, and architectural specifications are given in Appendix B and Appendix G. A preliminary notebook with code, models, and dataset is available in an anonymized repository: `https://anonymous.4open.science/r/mscd_verify-CE60`.

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

## A CROPROT CURATION DETAILS AND LIMITATIONS

**Inputs and pre-processing.** We use Sentinel-2 Level-2A surface reflectance (13 bands) resampled to 10 m over user-defined AOIs. Pairs $(T_1, T_2)$ are co-registered (projected to the same UTM zone), clipped to the AOI, and normalized consistently with the main pipeline (Fig. 1).

**Cloud/shadow screening and pair selection.** Candidates are filtered using a conservative cloud-probability threshold, with small morphological dilation to remove cloud fringes and shadows. We favor seasonal spacing that highlights crop-cycle differences while avoiding snow/flood outliers. Pairs with heavy haze, striping, or missing tiles are discarded.

**NDVI differencing and optional smoothing.** For each date, compute

$$\text{NDVI} = \frac{B_8 - B_4}{B_8 + B_4}, \qquad \Delta\text{NDVI} = \text{NDVI}_{T_2} - \text{NDVI}_{T_1}.$$

A small median filter (e.g., $3\times3$) may be applied to suppress speckle while retaining boundaries.

**Binarization and visual QA.** We derive a candidate threshold $\theta$ (e.g., Otsu on $\Delta$NDVI within a vegetation mask) and allow narrow manual adjustment to correct obvious BRDF/phenology artifacts. Each curated scene logs: AOI, product IDs/dates, cloud mask, final $\theta$, and a quick-look PNG (RGB and $\Delta$NDVI heatmap). Scenes failing sanity checks (e.g., residual clouds, strong BRDF seams) are excluded.

**Post-processing.** We apply small morphological operations (opening/closing with $3\times3$), drop connected components smaller than $s_{\min}$, and optionally enforce field-wise connectivity (8-neighbors). This yields the final binary proxy mask.

**Reproducibility.** All steps are scripted. The pipeline emits `.csv` manifests recording tile IDs, cloud statistics, thresholds, and paths to masks/previews, enabling exact re-generation.

**Limitations and intended use.** Masks are *weak supervision*: semi-automatic proxies, not hand-annotated GT. $\Delta$NDVI conflates phenology, BRDF, and mild misregistration; the QA step mitigates but cannot eliminate these effects. CropRot is therefore intended strictly as an *empirical OOD benchmark* for change detection, complementing OSCD. Certified robustness experiments in the main paper remain label-agnostic and probe bounded radiometric stability at the decision head.

## B VERIFIER DETAILS (HEAD-ONLY CERTIFICATION)

**Setup.** Given a trained change detector $f(x_1, x_2) = z \in \mathbb{R}^{2 \times H \times W}$ with channels $(c_{\text{chg}}, c_{\text{nchg}})$, we factor $f = T \circ B$ where the *body* $B$ is the encoder–decoder trunk up to the final skip concatenation and the *tail* $T$ is the last `DoubleConv` followed by the $1\times1$ head. Perturbations $(\delta_1, \delta_2) \in \Delta_\varepsilon$ are bounded by an $L_\infty$ radiometric budget $\varepsilon$ applied band-wise.

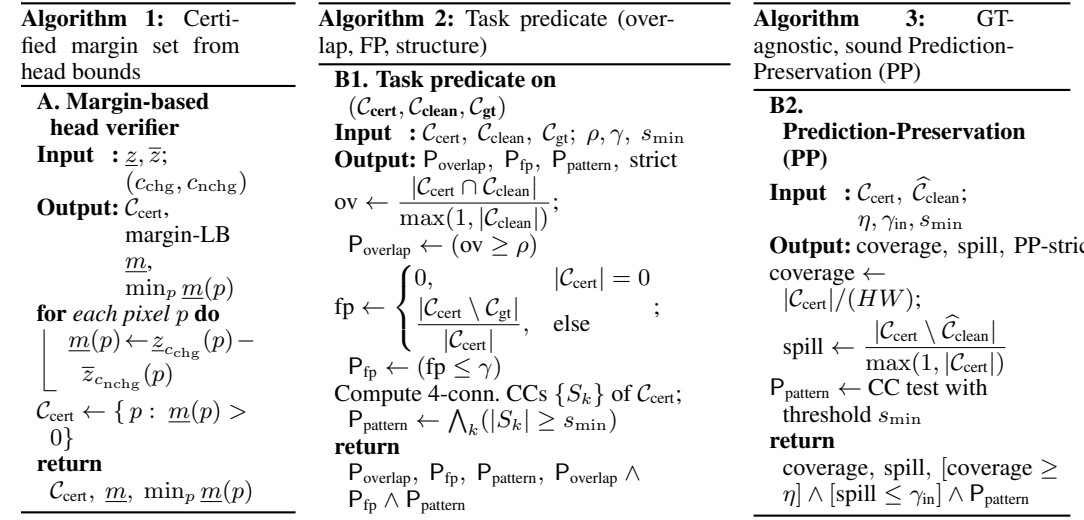

Figure 4: Algorithms for predicate verification.

**Interval transport to the tap.** We propagate the input box through $B$ using interval bound propagation (IBP) to obtain tap-domain bounds

$$z_t = B(x_1+\delta_1, x_2+\delta_2) \in [\ell, u] := [\underline{z}(B, \varepsilon), \overline{z}(B, \varepsilon)].$$

This step is cheap, architecture-agnostic, and summarizes upstream nonlinearity without attempting end-to-end relaxation.

**$\alpha$-CROWN on the tail.** On the short subnetwork $T$ (convs, biases, ReLUs/LeakyReLUs, and $1{\times}1$ head), $\alpha$-CROWN builds per-layer relaxations over $[\ell, u]$ and composes them to yield a global affine lower bound on the binary margin:

$$m_p = z_{c_{\text{chg}}}(p) - z_{c_{\text{nchg}}}(p), \qquad m_p\big(T(z_t)\big) \geq a_p^\top z_t + b_p, \ \forall z_t \in [\ell, u].$$

The certified margin is $m_p^{\text{lb}}(\ell, u) = \min_{z_t \in [\ell, u]}(a_p^\top z_t + b_p)$.

**Soundness. Theorem.** For all $(\delta_1, \delta_2) \in \Delta_\varepsilon$ and pixels $p$,

$$m_p(f(x_1+\delta_1, x_2+\delta_2)) \geq m_p^{\text{lb}}(\ell, u).$$

If $m_p^{\text{lb}}(\ell, u) > 0$, the class at $p$ is invariant to all admissible perturbations. *Sketch:* IBP guarantees $z_t \in [\ell, u]$; $\alpha$-CROWN provides a valid relaxation over $T$; taking the margin preserves soundness. $\square$

**Tightness vs. IBP-only. Proposition.** For IBP margin bound $m_p^{\text{IBP}}$, we have $m_p^{\text{lb}}(\ell, u) \geq m_p^{\text{IBP}}$. *Sketch:* interval composition is a special case of $\alpha$-CROWN's dual relaxation. $\square$

**Why widening intervals break certificates.** Writing $z_t = \frac{\ell+u}{2} + \xi, \xi \in [-\frac{u-\ell}{2}, \frac{u-\ell}{2}]$, yields

$$m_p^{\text{lb}}(\ell, u) \geq m_{p,\text{clean}} - \tfrac{1}{2}\sum_i |a_{p,i}|(u_i - \ell_i), \tag{2}$$

with $m_{p,\text{clean}} = a_p^\top \frac{\ell+u}{2} + b_p$. **Observation.** A pixel is certifiable only if $m_{p,\text{clean}} > W(p)$, where $W(p) = \frac{1}{2}\sum_i |a_{p,i}|(u_i - \ell_i)$. As $\varepsilon$ grows, $W$ increases due to widening tap intervals, driving coverage down even when clean margins look reasonable.

**Complexity.** If $k$ is the number of conv/activation layers in $T$ and $d$ the tap channels, complexity is $O(kHWd)$. Memory only stores per-layer coefficients, making it orders of magnitude lighter than end-to-end relaxations.

## B.1 GT-Aligned Predicates

Let $\mathcal{C}_{\text{cert}} = \{p : m_p^{\text{lb}}(\ell, u) > 0\}$ be the certified pixel set at radius $\varepsilon$. Let $\widehat{\mathcal{C}}$ be the clean change set and $\mathcal{C}^\star$ the ground truth.

- **Coverage:** $\text{cov}(\mathcal{C}_{\text{cert}}, \widehat{\mathcal{C}}) = \frac{|\mathcal{C}_{\text{cert}} \cap \widehat{\mathcal{C}}|}{\max(1, |\widehat{\mathcal{C}}|)}$. - **False Positives:** $\text{fp}(\mathcal{C}_{\text{cert}}, \mathcal{C}^\star) = 0$ if $|\mathcal{C}_{\text{cert}}| = 0$, else $\frac{|\mathcal{C}_{\text{cert}} \setminus \mathcal{C}^\star|}{|\mathcal{C}_{\text{cert}}|}$. - **Minimum Island Size:** all 4-connected components $S_k$ of $\mathcal{C}_{\text{cert}}$ satisfy $|S_k| \geq s_{\min}$.

We report the three metrics and optionally their strict conjunction: $[\text{cov} \geq \rho] \wedge [\text{fp} \leq \gamma] \wedge [\min_k |S_k| \geq s_{\min}]$.

## C IBP Baseline

A fully-IBP verifier (forward entire $f$) is scalable but looser: head bounds collapse even at $\varepsilon \in \{1, 2\}/255$. We retain IBP only as a diagnostic bound for $B$ and to generate $[\ell, u]$.

## D Architectures and Tap Placement

We verify FresUNet, FALCONet, and AttU-Net under the same contract: tap after the final decoder concat, before the last `DoubleConv`, with $T$ as that `DoubleConv`+OutConv. Tables 2–4 give complete layer specifications. All expose the same verifier interface.

## E Evaluation Details and Units

- **Perturbation radii:** $\varepsilon \in \{0, 1, 2\}/255$ on $[0, 1]$ inputs ($1/255 \approx 0.0039$). - **Preprocessing:** affine normalization transported through intervals avoids tightness errors at $\varepsilon = 0$.

## F Why Widening Encoder Intervals Break Certificates (Formal)

Let $m(z) \geq a^\top z + b$ for all $z \in [\ell, u]$. Writing around $z_0 = \frac{\ell + u}{2}$ gives

$$m_{\text{lb}}(\ell, u) \geq m_{\text{clean}} - W, \quad W = \tfrac{1}{2} \sum_i |a_i|(u_i - \ell_i).$$

**Corollary.** $\mathcal{C}_{\text{cert}} \subseteq \{p : m_{\text{clean}}(p) > W(p)\}$. Increasing width or sensitivity degrades predicate satisfaction.

## G Backbone Details and Verifier Interface

FresUNet, FALCONet, and AttU-Net all place the tap after Dec1 concat, before the final Double-Conv. Tables 2–4 list layer specifications. The verifier interface is identical across backbones.

**Conventions.** Inputs are two co-registered Sentinel-2 images stacked channel-wise (26 channels total: $2 \times 13$). All models output two per-pixel logits (change, no-change). A *DoubleConv* block denotes $\text{Conv}(3 \times 3) \rightarrow \text{BN} \rightarrow \text{ReLU} \rightarrow \text{Conv}(3 \times 3) \rightarrow \text{BN} \rightarrow \text{ReLU}$ with same padding. *Up* denotes bilinear upsample by $2 \times$ (or transposed conv, checkpoint-matched) followed by concatenation of the encoder skip and a *DoubleConv*. *OutConv* is a $1 \times 1$ convolution to two logits. Where attention is used, it is indicated explicitly. The verifier's *body* $B$ ends after the final skip concatenation; the *tail* $T$ is the last *DoubleConv* plus *OutConv*. The *tap* is inserted between $B$ and $T$.

—

## G.1 EncDec (vanilla U-Net style)

**Overview.** A symmetric encoder–decoder with four downsampling stages, a bottleneck, and four upsampling stages with skip concatenations; no attention modules.

Table 2: EncDec layer specification (channels shown as #feat at each resolution). Tap is placed after **Dec1 concat**, before the final **DoubleConv**.

| Stage | Resolution | #feat | Block | Notes |
|---|---|---|---|---|
| Enc1 | $H \times W$ | 64 | DoubleConv | input: 26 ch |
| Down1 | $H/2 \times W/2$ | 64 | MaxPool $2 \times 2$ | |
| Enc2 | $H/2 \times W/2$ | 128 | DoubleConv | |
| Down2 | $H/4 \times W/4$ | 128 | MaxPool $2 \times 2$ | |
| Enc3 | $H/4 \times W/4$ | 256 | DoubleConv | |
| Down3 | $H/8 \times W/8$ | 256 | MaxPool $2 \times 2$ | |
| Enc4 | $H/8 \times W/8$ | 512 | DoubleConv | |
| Down4 | $H/16 \times W/16$ | 512 | MaxPool $2 \times 2$ | |
| Bottleneck | $H/16 \times W/16$ | 1024 | DoubleConv | |
| Dec4 | $H/8 \times W/8$ | 512 | Up + Concat(Enc4) + DoubleConv | |
| Dec3 | $H/4 \times W/4$ | 256 | Up + Concat(Enc3) + DoubleConv | |
| Dec2 | $H/2 \times W/2$ | 128 | Up + Concat(Enc2) + DoubleConv | |
| Dec1 | $H \times W$ | 64 | Up + **Concat(Enc1) + DoubleConv** | **Tap after concat** |
| Head | $H \times W$ | 2 | OutConv $1 \times 1$ | logits: change/no-change |

**Verifier interface.** $B$ comprises Enc1→Dec1-concat (inclusive). $T$ is Dec1's *DoubleConv* plus *OutConv*. The tap exports interval bounds $[\ell, u]$ at the input of Dec1's *DoubleConv*.

—

### G.2  FALCONet (local convolutional attention)

**Overview.** A U-Net trunk augmented with lightweight convolutional attention (plus optional multi-head attention at selected scales) to refine local context; same skip and head interface as EncDec.

Table 3: FALCONet layer specification. Attention blocks appear after *DoubleConv* in the encoder and after concatenation in the decoder at mid/high resolutions. Tap is after **Dec1 concat**.

| Stage | Resolution | #feat | Block | Notes |
|---|---|---|---|---|
| Enc1 | $H \times W$ | 64 | DoubleConv + ConvAttn | input: 26 ch |
| Down1 | $H/2 \times W/2$ | 64 | MaxPool | |
| Enc2 | $H/2 \times W/2$ | 128 | DoubleConv + ConvAttn | |
| Down2 | $H/4 \times W/4$ | 128 | MaxPool | |
| Enc3 | $H/4 \times W/4$ | 256 | DoubleConv + ConvAttn | |
| Down3 | $H/8 \times W/8$ | 256 | MaxPool | |
| Enc4 | $H/8 \times W/8$ | 512 | DoubleConv + (MHA) | optional MHA |
| Down4 | $H/16 \times W/16$ | 512 | MaxPool | |
| Bottleneck | $H/16 \times W/16$ | 1024 | DoubleConv + (MHA) | optional MHA |
| Dec4 | $H/8 \times W/8$ | 512 | Up + Concat(Enc4) + ConvAttn + DoubleConv | |
| Dec3 | $H/4 \times W/4$ | 256 | Up + Concat(Enc3) + ConvAttn + DoubleConv | |
| Dec2 | $H/2 \times W/2$ | 128 | Up + Concat(Enc2) + ConvAttn + DoubleConv | |
| Dec1 | $H \times W$ | 64 | Up + **Concat(Enc1)** + ConvAttn + **DoubleConv** | **Tap after concat** |
| Head | $H \times W$ | 2 | OutConv $1 \times 1$ | logits: change/no-change |

**Verifier interface.** Same $B/T$ split as EncDec; the tap is placed after Dec1 concatenation, before its final *DoubleConv*. If attention wraps the head, the verifier descends to the first inner `Conv2d`.

—

## G.3 AttU-Net (attention-gated U-Net)

**Overview.** A U-Net in which each encoder feature is filtered by an attention gate before being concatenated with the decoder stream, suppressing irrelevant responses while preserving the standard head.

Table 4: AttU-Net layer specification. Attention gates (`AG`) modulate each skip before concatenation. Tap is after **Dec1 concat**.

| Stage | Resolution | #feat | Block | Notes |
|---|---|---|---|---|
| Enc1 | $H{\times}W$ | 64 | DoubleConv | input: 26 ch |
| Down1 | $H/2{\times}W/2$ | 64 | MaxPool | |
| Enc2 | $H/2{\times}W/2$ | 128 | DoubleConv | |
| Down2 | $H/4{\times}W/4$ | 128 | MaxPool | |
| Enc3 | $H/4{\times}W/4$ | 256 | DoubleConv | |
| Down3 | $H/8{\times}W/8$ | 256 | MaxPool | |
| Enc4 | $H/8{\times}W/8$ | 512 | DoubleConv | |
| Down4 | $H/16{\times}W/16$ | 512 | MaxPool | |
| Bottleneck | $H/16{\times}W/16$ | 1024 | DoubleConv | |
| Dec4 | $H/8{\times}W/8$ | 512 | Up + `AG`(Enc4) + Concat + DoubleConv | |
| Dec3 | $H/4{\times}W/4$ | 256 | Up + `AG`(Enc3) + Concat + DoubleConv | |
| Dec2 | $H/2{\times}W/2$ | 128 | Up + `AG`(Enc2) + Concat + DoubleConv | |
| Dec1 | $H{\times}W$ | 64 | Up + `AG`(Enc1) + **Concat + DoubleConv** | **Tap after concat** |
| Head | $H{\times}W$ | 2 | OutConv $1{\times}1$ | logits: change/no-change |

## H REPRODUCIBILITY

Scripts emit predicate logs (`predicate_pass_*.csv`) and LATEX tables. Knobs $(\rho, \gamma, s_{\min})$ are fixed unless noted; same tap across backbones; certification always on both dates jointly.

### H.1 IMPLEMENTATION NOTES AND GUARDS

- Uniform tap placement (post-Dec1 concat). - Pooling/upsampling semantics standardized. - Channel alignment checked before $\alpha$-CROWN. - GT masks sanitized to binary. - Safety guards: recursion limits, cached taps, stub rows (never false positives). - Bound normalization: per-channel affine standardization for stability. - Logging: clean margins, median widths, predicate summaries.

