# OpenReview forum: "Verifying Out-of-Distribution Robustness in Multi-spectral Satellite Change Detection"
_ICLR.cc/2026/Conference — ICLR 2026 Conference Withdrawn Submission_

### Official Review · Reviewer_J9Uo · 2025-10-29

**Soundness:** 2
**Presentation:** 2
**Contribution:** 2
**Rating:** 2
**Confidence:** 3

**Summary:**

This paper addresses the challenge of ensuring out-of-distribution (OOD) robustness for multi-spectral satellite change detection models, which is critical for their reliable deployment on-board satellites. The authors introduce two main contributions: Head-Consistency Training (HCT), a lightweight training objective that enforces stability at the model's decision head against physically-grounded perturbations like shadows and sensor drift, and a tail-tapped diagnostic verifier that applies $\alpha$-CROWN verification only to the final head to analyze why robustness fails. The work also contributes an OOD evaluation protocol: four Sentinel-2–motivated synthetic families plus CropRot, a curated vegetation-change benchmark built by ∆NDVI thresholding, both share OSCD’s 13-band interface while stressing different change phenomena. Experiments across three encoder–decoder backbones show that strict GT-aware certificates vanish at ε ≥ 1/255, even as HCT lifts clean OSCD performance and improves OOD Dice on CropRot. Diagnostics attribute certificate collapse to rapidly widening encoder intervals despite tight head spans.

**Strengths:**

1. The paper addresses a highly relevant and challenging problem of ensuring the robustness of multi-spectral change detection models. This is a critical need for real-world applications, especially for on-board satellite systems that must operate reliably in the face of test-time distribution shifts.
2. It introduces CropRot, a new, curated OOD benchmark specifically designed to test for vegetation-driven changes by using NDVI differencing with a visual quality assurance step . This dataset serves as an excellent semantic-shift benchmark to complement the urban-focused OSCD dataset, stressing models on different types of temporal dynamics.
3. The paper introduces Head-Consistency Training (HCT), a novel and practical method for improving empirical robustness. Unlike generic noise, HCT is physically grounded, enforcing stability against a set of four "sensor-calibrated" perturbations (low-frequency drift, shadowing, passband shift, and blur) that are motivated by real-world satellite imagery artifacts. HCT improves the empirical OOD robustness of the FALCONet model on the CropRot benchmark on the OSCD dataset.

**Weaknesses:**

1. Most results rely on OSCD (small, urban-change focus) and a curated CropRot set; the latter is explicitly geographically narrow and suggested for expansion across regions and seasons. The protocol is useful but not a strong OOD benchmark by itself.
2. The paper presents two main technical contributions: the tail-tapped verifier and the Head-Consistency Training (HCT) method. However, these two contributions are entirely disconnected. The paper explicitly states that the empirical gains from HCT do not translate into certified robustness and that HCT does not make models certifiably robust. The paper essentially presents a verification method that fails to certify a model trained with HCT.
3. The empirical results for HCT, while positive, are not overwhelmingly strong. The primary result is that HCT lifts a weaker backbone (FALCONet) to be more competitive. On the CropRot OOD benchmark, the baseline AttU-Net achieves a Dice score of 0.35. The proposed FALCONet with HCT only achieves 0.22. This suggests that architecture design (AttU-Net) is far more critical for OOD robustness than the proposed HCT training method.
4. There are minor concerns in section 3.2. For instance, the authors do not justify why $\tau$ was chosen from the range [0.05, 0.15]. Additionally, there is no ablation study for the $λ_{CE}$, $λ_{HCT}$, $λ_{Dice}$ values, and it is unclear what the $\sigma()$ function represents in the Dice loss.

**Questions:**

Please see the weakness.

---

### Official Review · Reviewer_KERw · 2025-11-01

**Soundness:** 2
**Presentation:** 2
**Contribution:** 2
**Rating:** 2
**Confidence:** 4

**Summary:**

This paper addresses the problem of out-of-distribution robustness in multi-spectral satellite change detection. It introduces a margin-based training strategy using physically motivated perturbations and a verifier that applies certified robustness analysis only to the model’s decision head. Experiments across several architectures demonstrate the difficulty of maintaining certification under realistic perturbations and identify the encoder–decoder body as the main source of instability. While the proposed training improves empirical robustness and transfer to new domains, these gains do not extend to certified robustness, which remains unverified for the proposed method.

**Strengths:**

1. Restricting alpha-CROWN verification to the head-only, tail-tapped computation is a clever and efficient design that offers valuable diagnostic insight into certification limits for satellite change detection models.

2. The paper tackles an important and timely problem, evaluating out-of-distribution robustness in change detection, and introduces two meaningful evaluation setups: the CropRot benchmark and physically motivated perturbations of OSCD.

3. The work is transparent in reporting negative results, such as the rapid collapse of certificates under small perturbations, and provides thoughtful diagnostic analysis to explain these outcomes rather than emphasizing benchmark gains alone.

4. The domain-specific design of both the CropRot dataset and the physically grounded perturbations enhances the paper’s practical relevance for satellite imagery applications, going beyond generic data augmentation techniques.

**Weaknesses:**

1. The architectural observations in Section 4.2.1 are insufficiently supported by experiments and are drawn from only three diverse architectures, limiting the strength of the conclusions.

2. The empirical value of HCT is weakly demonstrated: improvements are shown only for FALCONet, with limited comparisons against baselines and no ablation of perturbation types, standard augmentations, or adversarial training methods. Some non-HCT models even outperform FALCONet + HCT.

3. The tail-tapped verifier is not compared against any other verification approaches, leaving its impact on bound tightness relative to full end-to-end alpha-CROWN or other verifiers unclear.

4. The relationship between HCT and tail-tapped verification is not well established. The two methods appear conceptually independent, with little evidence that they complement or reinforce one another. Moreover, while the verifier provides useful diagnostic insights, these are not applied to analyze HCT-trained models.

5. The CropRot dataset relies on weakly supervised rather than manually curated labels, and the paper does not sufficiently justify or assess the reliability of these annotations.

**Questions:**

1. Clarify how HCT and tail-tapped verification are conceptually connected. If they are intended as complementary components of a unified robustness framework, emphasize this integration; otherwise, justify why they are presented together.
2. Include HCT-trained models in the diagnostic analyses to evaluate their certification behavior.
3. Evaluate HCT on additional architectures such as FresUNet and AttU-Net to assess generality across model types.
4. Clarify the rationale for using the perturbed OSCD benchmark given that all models collapse under these perturbations, and identify which perturbation types remain informative or realistic.
5. Report whether any architectural remedies identified through diagnostic analysis were implemented and tested experimentally.

---

### Official Review · Reviewer_LSTW · 2025-11-04

**Soundness:** 2
**Presentation:** 1
**Contribution:** 2
**Rating:** 2
**Confidence:** 4

**Summary:**

The paper aims to explore the question of how robust are trained change detection models for deployment on-board of satellites and proposes a methodology to study this and potentially improve the robustness. Their approach ultimately adds a loss that punishes if the model predictions deviate too much from the ground truth label given some perturbations. These perturbations are further somewhat inspired by the domain in quesiton, remote sensing data on-board of satellites - by using a change detection dataset made of Sentinel-2 images.
The authors of the paper plan to release their code and also a small dataset made in this paper to further examine the out-of-distribution robustness of trained models. The created dataset aims to target plant growth changes as they occur over seasons.
There are couple of good points about this paper, but also quite severe weaknesses - I will go into more details below.

**Strengths:**

The concept of this paper is relatively solid: indeed for machine learning models trained on-ground, there is a rigorous process needed before they can be deployed on-board of satellites. However, I would say, that there are quite some problems in framing the paper in this domain, without actually citing almost anything from either Remote Sensing literature, or the more specialised AI on-board (of satellites) literature.

Secondly, it is appreciated, that the authors are planning to release their code and the data they collected.

**Weaknesses:**

Jargon
It needs to be noted, that the paper starts with quite heavy jargon and doesn't define or explain the used terms very well. Also, it doesn't start with a general description of the methods and topics before going into the details and names of methods. There are several points where this could be done, either in the introduction, or in the background literature. (For comparison, for example one of the already cited papers "Certified Adversarial Robustness via Randomized Smoothing" does this better).

Language
This point is somewhat connected to the heavy use of jargon, but goes beyond. The language used in this paper is somewhat strange and doesn't really match terminology used in machine learning literature. Some terms are left without definitions and it is upon the reader to try to figure out what these mean. (An incomplete list of confusing terms is attached bellow) Maybe the use of LLMs was to the detrement of this paper (if it's the LLMs that introduce this terminology).

Dataset
The paper presents a new dataset made from pairs of Sentinel-2 images and mostly automatically created change labels based on NDVI. This on its own might not be very reliable - the paper describes these limitations, but only in the appendix - it would be suggested to keep this discusion about these limitations inside the paper. Details about the dataset are also missing - there is no number of scenes, their extend, their geographical distribution around the world (typically in papers that present datasets, there would be a map showing this distribution). The dataset seems to be quite small, similarly to the OSCD dataset, which also has just a few labelled scenes. For these reasons, its difficult to estimate how well models generalise on these datasets altogether. Similarly, models trained on these likely won't be robust to any variance of data (here presented OOD perturbations, or even scenes from different location).

Domains of Remote Sensing and AI On-board Satellites
Despite using problems from these two domains as the backdrop of the paper, there is a large lack of almost any literature being cited from these two disciplines (and their intesection) (except some of the work by R.C Daudt and few other classical papers). It could be argued that this would be beyond the paper, but there needs to be a more details and knowledge cited from these disciplines. For example, the used perturbations were seemingly inspired from Sentinel-2 data - no citation is provided to any works that would encounter these. Also no references to how the multispectral data tends to be pre-processed before using by ML models. There are works that propose either use of ML models directly on-board of satellites, and explicitely address the problems of deployment with apriori unknown characteristics of the later used sensors (see "Towards global flood mapping onboard low cost satellites with machine learning" and "In-orbit demonstration of a re-trainable machine learning payload for processing optical imagery" both by Gonzalo Mateo-Garcia et. al.). There are also works that address change detection on-board of satellites and were later deployed in real space missions ("RaVÆn: unsupervised change detection of extreme events using ML on-board satellites" by Vit Ruzicka et al). In general, the task of change detection doesn't have to be solved by encoder-decoder networks, in fact the mentioned papers use unsupervised learning approaches that may be better for dealing with sensor degradation and OOD data at the time of model deployment.
Why this matters, is because this paper is in it's first sentence specifying that it focuses on multispectral change detection on-board of satellites, while it doesn't really go into the details that would arise in these situations. One example is that getting correctly co-registered data on-board might be very difficult (read slow to compute, hard to store on-board in limited storage etc.) - for example for that reason, a sensible perturbation might be misalignment of input tiles.


Results
The formatting of the results in the table 1 could be improved - list of models without the proposed HCT technique is around the one model that uses it. Figure 3 doesn't have labels for classes (and also the shown predictions look like the models have not learned much, if I understand it correctly, the FALCONet is predicting "change" everywhere - this seems like a training collapse). The terms used in the diagnostics table (1a) also seem quite confusing to me. Finally, some section in the results aren't formatted as full text paragraphs, instead they are bullet points, this makes it hard to read (especially when the points are not really expanded on).

Confusing terms (mentioned in the point about language):
- "end-to-end relaxations quickly become vacuous"
- "not adversarial and not verifier-coupled"
- tap bounds
- head relaxations
- shallow-body verification
- "inside the graph"
- predicate outcomes
- Head logit spans / Head logit span / head spans remain tight
- Tap widths explode / Tap width_mean
- keep tap boxes tight and head sensitivity
- architecturally narrow the tap interval
- AttU-Net "transfers" best (maybe generalises would be better?)
- bottleneck lies in the body
- The pipeline emits .csv manifests


-----
Smaller details:

APPENDIX
The paper sends the reader to the appendix quite a lot, it would be better to provide more explanations in the main text of the paper instead.

ABBREVS
Some abbreviations were not expanded properly:
change detection (CD)
visual quality assurance (VQA)

**Questions:**

-

---

### Official Review · Reviewer_MuUM · 2025-11-04

**Soundness:** 2
**Presentation:** 1
**Contribution:** 2
**Rating:** 2
**Confidence:** 3

**Summary:**

This paper investigates robustness of multi-spectral satellite change detection on both certification and empirical perspectives.

**Strengths:**

Satellite change detection seems an interesting application.

**Weaknesses:**

1. unclear academic novelty
2. unclear writing that is really hard to follow: many acronyms without explanations, missing references, lack of description on satellite change detection.

**Questions:**

N/A

---

### Note · Authors · 2025-11-12

**Comment:**

We sincerely thank the reviewers for their constructive feedback and valuable suggestions. After careful consideration, we have decided to withdraw this submission and plan to resubmit a revised version to a venue more focused on formal verification.

**Withdrawal Confirmation:**

I have read and agree with the venue's withdrawal policy on behalf of myself and my co-authors.